# What determines participation in sport for older adults in England: A multilevel analysis of Active Lives data

**Andrew Brinkley** [iD]*, **Gavin Sandercock** [iD], **Ruth Lowry** [iD], **Paul Freeman**

School of Sport, Rehabilitation and Exercise Sciences, University of Essex, Essex, United Kingdom

* andrew.brinkley@essex.ac.uk

## Abstract

Physical inactivity within an ageing population is an ongoing public health concern for policy-makers. Engagement in sport forms a foundation of policy designed to encourage physical activity participation and improve health and wellbeing. This study aimed to (i) understand the extent to which older adults participate in sport and the (ii) correlates that predict this involvement within an English population sample of older adults. A further aim was (iii) to examine the extent in which sports participation may vary due to the opportunity provided across Active Partnerships in England. To address this, a multi-level analysis framed through COM-B was conducted of the 2021 English Active Lives dataset (i.e., during the COVID-19 pandemic). The Active Lives survey provides population-level insight into sport, exercise, and physical activity participation across England. It samples upwards of $n = 180,000$ participants beyond the age of 16 years and asks questions on factors that influence participation. Our findings drawn from a sample of $n = 68,808$ older adults (i.e., >60-years of age) indicate that when accounting for variation across regions sports participation was significantly predicted by age ($\beta = -.246$, $p = .040$) and multiple deprivation ($\beta = .706$, $p = .030$). Further, our analysis suggests sports participation across regions is associated with changes in the perceptions of opportunity to participate ($\beta = -28.70$, $p = .001$). As the UK transitions from the COVID-19 pandemic, findings have implications for the promotion of sports participation for older adults, in that local, regional, and national stakeholders must do more to change perceptions of social and physical opportunity within an ageing population. This may be achieved through adaptations to the recreational sporting landscape, raising awareness, and supportive policy changes on a national level.

**Data Availability Statement:** All data is available from the UK Data Archive (http://doi.org/10.5255/UKDA-SN-8993-1).

## Background

Populations globally are ageing and becoming more inactive [1, 2]. Physical activity participation is a modifiable risk factor for non-communicable disease and illness [3], the effect of relative ageing [4], and mental health, cognitive function and wellbeing [5, 6] in older adults. Participation in regular physical activity can improve physical function and prevent falls [2, 6], and if conducted in groups promote social health outcomes, such as cohesion, friendship, and

**Funding:** The author(s) received no specific funding for this work.

**Competing interests:** The authors have declared that no competing interests exist.

**Abbreviations:** COM-B, Capability, Opportunity, Motivation–Behaviour; CAWI, Computer Assisted Web Interviews; MEMS, Moderate Equivalent Minutes; MITS, Moderate Intensity Minutes; VITS, Vigorous Intensity Traditional Sports; BMI, Body Mass Index; ICC, Interclass Correlation.

belonging [7]. Within England, despite minor increases in participation since 2015, in 2021, 38.5% of 55–64 years old, 39.9% of 65–74 years old, 56.9% of 75–84 year old, and 78.4% of adults over the age of 84 years [8] did not participate in 150 minutes of moderate intensity physical activity per-week (i.e., one component of the national physical activity guidelines for older adults) [9]. These data remain consistent with other high-income countries globally [10]. This places an increased burden on health and social care, and public health resources and capacity [1, 2, 11].

The COVID-19 pandemic has influenced population-level health, wellbeing, and quality of life [12, 13] and physical activity behaviour [8, 14]. Further, the impact of the pandemic has shaped the political, economic, societal, and policy landscape, with notable reduced public spending as countries transition into an endemic phase [12, 13]. Factors during the pandemic, as a response to this, and as a function of its lasting impact, such as home-quarantine and self-isolation [15], closure or adaption of facilities [16, 17], and regional and national mitigation strategies [18] may shape an individual's ability to participate in physical activity.

Sport England (i.e., a non-executive public body of the UK Government) are tasked with developing, evaluating, and implementing programmes and policies to promote participation in recreational physical activity (including exercise and sport) [19]. After the 2012 London Olympic Games, which were envisaged to 'inspire a generation,' Sport England and the UK Government have implemented policies and strategies such as 'A Sporting Habit for Life' (2012–2017), 'Towards an Active Nation' (2016–2021), 'Uniting the Movement' (2021–2030), and more recently, 'Get Active' (2023–2030). As these policies and strategies have evolved, a greater emphasis has been placed on (i) recreational physical activity including sport participation, (ii) place-based solutions within regions, (iii) understanding the correlates of physical activity behaviour, and (iv) implementing multi-level systems change [19]. Within the past six years, Sport England has invested upwards of £1.9 billion (i.e., £323 million per-annum) on targeting inactivity nationwide [20].

To facilitate participation in sport, exercise, and movement, Sport England invests in 43 Active Partnerships (e.g., regions, cities, or counties). Active Partnerships are independent organisations who work with local, regional, and national stakeholders (e.g., commercial, education, healthcare and community organisations, governments, charities, public health partners) to coordinate, create and support social and physical opportunities (e.g., changing how individuals' access and participate) for the delivery of sport, exercise, and movement in England. This is achieved through the delivery of programmes, connecting stakeholders, sports development, insight, funding, and raising awareness. This coordination, delivery and sustainability is set against the backdrop of a complex systems of interacting intrapersonal-, interpersonal-, environmental- and policy-level correlates [19, 21]. Therefore, how each Active Partnership implements or works with systems-wide stakeholders, partners and deliverers to opportunity varies. For example, this may not be limited to the provision of new facilities, promotion of existing clubs and provisions, or strategies designed to address situational place-based health complications. Given a decade of UK Government policy, which outlines the provision of sport and exercise should be equal for all [19], understanding what determines participation across Active Partnerships (i.e., across regions) and within modes of activity is vital to ensure the long-term sustainability of delivery within England [21].

Sport is conceptualised as a competitive/non-competitive and informal/formal mode of physical activity that follows some form of pre-established structured rules, structure and tradition [22]. Sport is a hallmark of UK health promotion policy [19], and there is strong evidence to indicate over the short-, medium-, and long-term, participation in a range of sports (e.g., football, walking netball, basketball) can improve physical (i.e., musculoskeletal, physical function, cardiorespiratory, cardiometabolic and body composition) [4, 23–26], psychological

(i.e., mental health, wellbeing, quality of life) [27], and social health outcomes (i.e., reduced loneliness and social isolation) [4, 26, 28] in older adults. Further, evidence drawn from systematic reviews of localised studies on sports participation (e.g., a given region, sport or programme) [25, 26, 29–31] in older adults and existing analysis of Active Lives physical activity data [14, 32–34] shows that participation in sport is determined by socio-geographical factors (i.e., age, employment status, ethnicity, gender, occupation, living status, social-economic status, marital status and activity opportunities) and individual-level psychological factors (e.g., perceived health capability, and motivation). This underscores the importance of charting the behavioural correlates that underpin sports participation in a nationwide population of older adults across England.

The meta-behavioural theory known as COM-B [35] is an accepted lens to frame behaviour within research surrounding sports and physical activity participation [36–38], UK Governmental policy [19], and the Active Lives survey [8, 39]. The model is based primarily on three interacting psychological factors (Capability, Opportunity, and Motivation). Within COM-B, capability is the degree in which an individual perceives they have physical and psychological ability to participate in an activity. Opportunity is the range in which an individual has the physical and social prospect to participate. Finally, motivation represents the extent to which automatic habits or reflective thoughts drive behaviour [35]. Whilst influenced by social and environmental factors, capability and motivation are individual-level predictors of behaviour within COM-B [35]. In contrast, opportunity varies according to social support, subjective norms, and environmental factors such as the availability of facilities and/or social actors to support participation [35]. As such, opportunity can, in theory, be influenced to a meaningful extent by the relationships an Active Partnership builds with regional and local governmental policymakers, such as the integrated care system, healthcare sector, sports and leisure providers, charities and advocacy groups, and education establishments.

Previous research using COM-B to understand older adults' participation in sport is sparse [40]. Two previous qualitative studies [31, 40], that aimed to understand walking football participation, and a recent meta-analysis of 'exergaming' [41] involvement in older adults found physical and psychological capability (e.g., existing health and experience), automatic motivation (e.g., integration and identification with sport), and social (e.g., supportive and resourced clubs) and physical opportunity (e.g., lack of suitable facilities or sessions) to be modifiable correlates of participation. These studies emphasised the importance of adaptable facilities and supportive social settings to encourage participation for older adults [31, 40, 41]. These are components of opportunity that local and regional authorities, organisations, and bodies such as Active Partnerships hold meaningful responsibility for [19].

Although these reviews [25, 26] and studies [14, 31–34, 40] provide a strong foundation, the participation of older adults within sport is seldom understood on a population-level. For example, what is known about the factors that predict participation in sport for older adults is limited to process-evaluations of individual programmes, data drawn from individual sports [31, 40], regional or non-population level cross-sectional studies, and small qualitative studies [21, 25, 26]. Further, where analysis has been conducted on population samples [14, 32–34], this is limited to broader predictors of physical activity, which may differ to that of sports participation. Importantly, research has also not considered how correlates of behaviour are associated across regions of a nationwide population. Given the systems-thinking ambitions of Sport England and the importance of 'place-based solutions' within Active Partnerships [42], this underscores the need to examine the variation explained by a range of individual- and regional-level predictors within a population-level sample [32–34]. Addressing this challenge may better allocate the funding and resources of the Government, Sport England, and Active Partnerships, and reduce potential regional inequalities in sports participation.

### Aim

Through a multi-level modelling analysis and accounting for variation across Active Partnerships, the aim of the present study was to (i) understand the extent to which older adults participate in sport and the (ii) correlates that predict this involvement within an English population sample of older adults. A further aim was (iii) to examine the extent in which sports participation may vary due to the opportunity provided across Active Partnerships in England.

## Methodology

### Design and sampling

These data were drawn from cross-sectional data of the November 2020–2021 (Year 6) Active Lives survey [34, 39]. These data were collected during the COVID-19 pandemic, which included one nationwide lockdown and varying periods of regional social restrictions and personal mitigation behaviours. The Active Lives survey is an English nationwide population survey of physical activity behaviour [34, 39] delivered by IPSOS MORI (i.e., a UK based market research company). The purpose of the survey is to recruit a population representative sample (n = 180,000 participants beyond the age of 16 years) across regions (i.e., minimum 500 participants per local authority). Within the 2020–2021 survey, n = 177,273 participants were sampled through random probability sampling of the Royal Mail's Postcode Address File. To account for seasonal variation and impact of the COVID-19 pandemic, the Active Lives survey is distributed through 12-monthly waves of push-to-web computer assisted web interviews (CAWI) or paper questionnaires. Sampling procedures are outlined in detail via Sport England [43]. The overall response rate of the survey was 22.5%. Data were sought from the UK Data Archive (data and technical methodologies are available here) [43]. Ethical approval was not required to undertake the analysis of secondary data. Given this study focused on older adults, participants under 60 years of age or with missing age data were removed from the analysis (n = 107,430). Further, n = 101 participants without basic demographic data (e.g., gender) or with miscoded responses were removed from the analysis. Upon removal, n = 69,742 were available for an initial analysis. Thresholds for self-report physical activity indicate a maximum cut-off of >3360 minutes ((8*60 minutes)*7 days) [34, 44]. Therefore, data reporting >3360 minutes of traditional sport per week were removed from analysis (n = 934). A final sample of n = 68,808 participants were eligible for final analysis. Within the final sample, 54.8% participants completed the survey via CAWI and 45.2% through a paper survey. To assess the need for a weighted analysis, comparisons were drawn against 2021 general population census data for stratified age, gender, ethnicity and multiple deprivation [45]. Based on these data, the data did not warrant weighting for population representativeness. IPSOS MORI gained informed consent from each participant. The study processes are consistent with the Declaration of Helsinki [46].

### Measures

**Sports participation.** The Active Lives survey requested participants to provide data on the number of days they had participated in 200 different modes (limited to 50 activities for the paper version) of physical activity, including sport (a full list of available sports is available via the UK Data Archive), in the last 28-days (4-weeks) [34, 39]. If a participant had participated in an activity, they were subsequently requested to provide data on the duration (hours and minutes in a typical session) and intensity ('enough to raise your breathing rate'—moderate intensity and 'enough to make you out of breath or sweat'–vigorous intensity) of their participation. Sport, exercise, and forms of leisure-time activity were transposed by IPSOS MORI

as a series of sports and modes of exercise composite variables (activity classifications are available on the UK Data Archive) [34, 39]. For the purposes of the present study, Moderate Equivalent Minutes (MEMS) spent participating in traditional modes of sport was extracted from the data. MEMS traditional sport is formed of activities representing moderate- (MITS) and vigorous-intensity traditional sports (VITS) into a single composite variable. MITS and VITS are created by IPSOS MORI by multiplying the number of bouts of each reported activity in the past 28-days by either one for MITS or two for VITS. This 28-day variable is then divided by four to provide a weekly estimate of traditional sports participation. A comprehensive overview of variable derivation, data cleaning and methods to minimise double-counting of activities is available on the UK Data Archive (see http://doi.org/10.5255/UKDA-SN-8993-1) [39].

**Individual-level correlates of behaviour.** Individual variables extracted from the Active Lives Survey included self-reported age, body mass index (kg/m$^2$) (BMI), gender, disability status, education, ethnicity, living arrangements, socioeconomic classification (higher and middle social occupations, lower occupational, students/unclassified) [47], self-reported health status (1 = very good– 5 = very bad), and employment status. Variables representing elements of the COM-B model were also extracted from the data. We extracted a single item for amotivation, external, introjected, identified and intrinsic motivation regulation (this form of motivation represents me; 1 = agree– 5 = disagree), which are from the Behavioural Regulations in Exercise Questionnaire 3 [48]. Capability was represented by the individual's perceived physical and psychological ability to participate in physical activity or sport (i.e., I feel that I have the ability to be physically active. . ..in relation to sports, fitness and recreational activities; 1 = strongly agree– 5 = strongly disagree) [35]. Generalised self-efficacy (termed individual development within the Active Lives Survey) (i.e., I can achieve most goals I set myself; 1 = strongly disagree– 5 = strongly agree) was also extracted from the data. The four ONS measures of wellbeing, namely, anxiety (i.e., how anxious did you feel yesterday), happiness (subjective wellbeing) (i.e., how happy did you feel yesterday), worthwhileness (i.e., to what extent do you feel the things you do in your life are worthwhile), and life satisfaction (i.e., how satisfied with your life right now) were also extracted from the data. These items were assessed on a 0–10 scale, where by 0 represented not accurate representation, and 10 a complete representation. Finally, we extracted a single item for loneliness from the data (i.e., how often do you feel lonely; 1 = often– 5 = never).

**Active partnership-level correlates of behaviour.** Variables extracted on the Active Partnership level were the 2019 index of multiple deprivation total deciles (1–10; 1 = most deprived—10 = least deprived) and rurality (proxy measures from postcode), opportunity to participate in sport, exercise, and physical activity (i.e., I feel that I have the opportunity to be physically active. . ..in relation to sports, fitness and recreational activities) (1 = strongly agree– 5 = strongly disagree) [35], community needs index (i.e., active/engaging community, civic assets, connectedness), number of clubs attended, and local trust (i.e., to what extent do you agree or disagree: that most people in your local area can be trusted; 1 = strongly disagree– 5 = strongly agree).

## Data analysis

Data were analysed through MLwiN Version 3.05. Descriptive statistics and bivariate correlations were calculated for continuous variables on the participant level. Frequencies (i.e., gender, disability status, education, ethnicity, living arrangements, socioeconomic classification) were calculated for categorical variables. Data were analysed using a two-level multilevel regression model, where participants (Level 1) were nested into their local authority Active Partnership (Level 2). Multilevel models assess interindividual variability within hierarchal

and clustered structures (e.g., across regions) [49]. These models are useful in analysing equivocal data (e.g., dependent error structures, unequally spaced data collection, missing data, heterogenous variance, non-normally distributed data, moderation effects, and time-based covariance) common with population data collection [49]. The model was estimated through Iterative Generalised Least Squares and sequentially constructed. We constructed a variance component only (null) model to establish the interclass correlation coefficient (ICC). Secondly, a random-intercept model was constructed whereby fixed grand mean centred explanatory variables were entered into the model on the individual-level. Variables were selected based on previous research [14, 25, 26, 32–34]. More specifically, age, gender, disability, BMI, multiple deprivation, capability, opportunity, intrinsic, identified, and amotivated motivational regulations, health status, happiness and self-efficacy, loneliness, and community needs index were entered into the model. To explore how the relationship between explanatory variables and sports participation across Active Partnerships a random slope (model 2) was constructed. At each stage model fit was calculated through 2*loglikelihood and $\chi^2$ distribution tests for significance, and unstandardized coefficients were used.

## Results

### Descriptive statistics and bivariant correlations

Unadjusted analysis indicates participants played 68.27±222.30 MEMS of traditional sport per-week. Participants (51% female) were in the majority white British (92.2%), aged 70.67 ±7.49 years, and reported a BMI of 26.38±4.83. Most participants (41%) were educated at or beyond a degree level (level 4) and lived in a coupled household (58.7%). Participants (24.7% disabled) were in the majority, retired (74.5%). These participants were from higher to middle class social occupational groups (72.65%); however, 35.9% lived below the mid-point of multiple deprivation. Most of the participants sampled were from urban areas (72.4%). Participants were nested within n = 43 Active Partnerships. Mean MEMS of traditional sport is reported within Table 1. Descriptive data and bivariate correlations are presented within Table 2.

### What predicts sports participation in older adults?

The variance component model was a significantly better fit of the data than a general linear level regression model ($\chi^2$ 1, n = 68,808) = 907596.697, $p$ = .001. A strong interclass correlation indicated (ICC = .82) indicated the needs for a multilevel analysis [50]. Model 1 (random-intercept model) explained 47% of the variance in the outcome. The random-intercept and slope model (model 2) allowed sports participation at the Active Partnership level to vary as a function of changes within opportunity to play. This model explained 53% of the variance in the outcome and was a good fit of the data ($\chi^2_{df\ =\ 20}$, $p$ = .001). Model 2 indicated the random slopes of MEMS sports participation varied significantly ($p$ = .001) due to differences in opportunity across Active Partnerships. More specifically, as opportunity was perceived as less favourably, sports participation decreased meaningfully ($\beta$ = -28.70, $p$ = .001). Moreover, analysis of individual level predictors indicates MEMS of sports participation was significantly predicted by age ($\beta$ = -.246, $p$ = .040) and multiple deprivation ($\beta$ = .706, $p$ = .030). All other predictors were non-significant. Model fit statistics and parameter estimates are available in Table 3.

## Discussion

The present study provided a population-level analysis of the sports participation of older adults residing in England at various stages of the COVID-19 pandemic. Previous research

**Table 1. Traditional sport moderate equivalent minutes and demographic data.**

| Active Partnership (*n* = participants within region) | Traditional Sport Moderate Equivalent Minutes Per-Week (Mean±Standard Deviation) | 95% Confidence Interval | |
|---|---|---|---|
| | | Lower Bound | Upper Bound |
| Bedfordshire and Luton (*n* = 508) | 59.74±215.88 | 40.92 | 78.56 |
| Berkshire (*n* = 981) | 66.68±220.67 | 52.85 | 80.50 |
| Birmingham (*n* = 560) | 46.83±144.59 | 34.83 | 58.83 |
| Black Country (*n* = 742) | 44.98±172.47 | 32.55 | 57.41 |
| Buckinghamshire and Milton Keynes (*n* = 403) | 79.33±234.81 | 56.33 | 102.32 |
| Peterborough and Cambridgeshire (*n* = 1100) | 59.61±210.33 | 47.17 | 72.06 |
| Cheshire (*n* = 620) | 73.33±214.95 | 56.78 | 90.68 |
| Cornwall and Isle of Scilly (*n* = 335) | 70.40±197.28 | 49.20 | 91.60 |
| Cumbria (*n* = 1374) | 144.29±344.05 | 126.08 | 162.49 |
| Derbyshire (*n* = 1839) | 96.68±264.52 | 84.55 | 108.80 |
| Devon (*n* = 2456) | 90.67±257.56 | 80.48 | 100.87 |
| Dorset (*n* = 492) | 83.64±254.26 | 61.12 | 106.16 |
| Durham (*n* = 225) | 92.72±266.34 | 57.73 | 127.71 |
| Essex (*n* = 2895) | 54.24±199.77 | 46.96 | 61.52 |
| Gloucestershire (*n* = 1287) | 76.59±234.82 | 63.75 | 89.43 |
| Manchester (*n* = 3770) | 70.27±219.61 | 63.26 | 77.29 |
| Hampshire and the Isle of Wright (*n* = 2845) | 69.47±228.84 | 61.06 | 77.88 |
| Herefordshire and Worcestershire (*n* = 1607) | 73.22±221.35 | 62.39 | 84.05 |
| Hertfordshire (*n* = 1807) | 66.68±213.34 | 56.84 | 76.53 |
| Humberside (*n* = 821) | 50.49±182.94 | 37.96 | 63.03 |
| Kent (*n* = 2730) | 64.73±234.69 | 55.92 | 73.54 |
| Lancashire (*n* = 2895) | 75.53±233.66 | 67.02 | 84.05 |
| Leicestershire and Rutland (*n* = 1875) | 67.33±227.47 | 57.03 | 77.64 |
| Lincolnshire (*n* = 1544) | 46.73±182.83 | 37.61 | 55.86 |
| London (*n* = 4314) | 56.48±184.75 | 50.97 | 62.00 |
| Merseyside (*n* = 1676) | 48.89±167.81 | 40.85 | 56.93 |
| Norfolk (*n* = 1656) | 50.17±188.50 | 41.09 | 59.26 |
| North Yorkshire (*n* = 1804) | 105.69±292.06 | 92.20 | 119.17 |
| Northamptonshire (*n* = 1375) | 51.13±165.32 | 42.38 | 59.87 |
| Northumberland (*n* = 252) | 94.87±240.18 | 65.07 | 124.67 |
| Nottinghamshire (*n* = 1967) | 65.22±217.90 | 55.58 | 74.85 |
| Oxfordshire (*n* = 992) | 74.17±231.91 | 59.72 | 88.62 |
| Shropshire (*n* = 459) | 71.89±231.85 | 50.62 | 93.15 |
| Somerset (*n* = 893) | 69.90±225.55 | 55.09 | 84.72 |
| South Yorkshire (*n* = 2513) | 56.96±201.88 | 49.07 | 64.86 |
| Staffordshire (*n* = 1895) | 59.73±217.34 | 49.94 | 69.52 |
| Suffolk (*n* = 1128) | 56.28±211.12 | 43.94 | 68.61 |
| Surrey (*n* = 2087) | 85.11±241.46 | 74.74 | 95.47 |
| Sussex (*n* = 3059) | 69.52±229.09 | 61.39 | 77.64 |
| Tees Valley (*n* = 1002) | 55.87±209.41 | 42.89 | 68.86 |
| Tyne and Wear (*n* = 1398) | 54.35±207.11 | 43.49 | 65.22 |
| Warwickshire (*n* = 1473) | 51.91±181.76 | 42.62 | 61.20 |
| West of England (*n* = 1106) | 61.18±190.12 | 49.96 | 72.40 |
| West Yorkshire (*n* = 1685) | 70.35±218.67 | 59.91 | 80.80 |
| Wiltshire and Swindon (*n* = 372) | 76.84±264.85 | 49.84 | 103.84 |

**Table 2. Descriptive statistics and bivariant correlations.**

| Variable | M±SD | r^(p) | | | | | | | | | | | | | | | | | | | |
|---|---|---|---|---|---|---|---|---|---|---|---|---|---|---|---|---|---|---|---|---|---|
| | | 1 | 2 | 3 | 4 | 5 | 6 | 7 | 8 | 9 | 10 | 11 | 12 | 13 | 14 | 15 | 16 | 17 | 18 | 19 | 20 |
| 1. MTS | 68.27±222.30 | - | | | | | | | | | | | | | | | | | | | |
| 2. Age | 70.67±7.49 | -.10* | - | | | | | | | | | | | | | | | | | | |
| 3. BMI | 26.38±4.83 | -.07* | -.09* | - | | | | | | | | | | | | | | | | | |
| 4. HeStat | 2.30±.90 | -.15* | .15* | .26* | - | | | | | | | | | | | | | | | | |
| 5. IM | 2.18±.99 | -.19* | .07* | .23* | .40* | - | | | | | | | | | | | | | | | |
| 6. IDM | 1.98±.89 | -.15* | .06* | .18* | .31* | .69* | - | | | | | | | | | | | | | | |
| 7. IJM | 2.69±1.07 | -.07* | .05* | .06* | .15* | .39* | .50* | - | | | | | | | | | | | | | |
| 8. EM | 3.89±.87 | .02* | -.07* | -.06* | -.03* | -.02* | .04* | .24* | - | | | | | | | | | | | | |
| 9. AM | 3.61±.82 | .05* | -.03* | -.06* | -.12* | -.29* | -.21* | -.14* | -.00 | - | | | | | | | | | | | |
| 10. Capab | 2.15±1.07 | -.19* | .23* | .26* | .63* | .56* | .46* | .26* | -.00 | -.14* | - | | | | | | | | | | |
| 11. Anxiety | 2.78±2.74 | -.05* | -.01 | .01 | .25* | .12* | .06* | -.05* | -.05* | -.05 | .18* | - | | | | | | | | | |
| 12. Happy | 7.40±2.07 | .08* | .00 | -.08* | -.38* | -.22* | -.15* | -.03* | -.02 | .12* | -.29* | -.45* | - | | | | | | | | |
| 13. SelEff | 3.72±.78 | .10* | -.09* | -.10* | -.41* | -.29* | -.21* | -.11* | -.01 | .09* | -.40* | -.28* | .42* | - | | | | | | | |
| 14. LifeSat | 7.25±2.04 | .09* | -.01 | -.10* | -.41* | -.23* | -.16* | -.04* | -.03* | .12* | -.33* | -.38* | -.84* | -.44* | - | | | | | | |
| 15. Lonely | 3.71±1.19 | .06* | -.02* | -.05* | -.25* | -.11* | -.06* | .04* | .04* | .06* | -.18* | -.38* | .51* | .30* | .50* | - | | | | | |
| 16. IMD | 6.41±2.66 | .07* | .05* | -.11* | -.15* | -.09* | -.09* | -.06* | .00 | .06* | -.12* | -.04* | .07* | .06* | .08* | .09* | - | | | | |
| 17. Opp | 1.95±.92 | -.17* | .19* | .24* | .56* | .50* | .41* | .25* | .00 | -.17* | .78* | .17* | -.28* | -.38* | -.31* | -.20* | -.14* | - | | | |
| 18. CNI | 68.33±36.59 | -.05* | -.02* | .10* | .08* | .05* | .06* | .04* | -.01 | .04 | .08* | .00 | -.01 | -.03* | -.02 | -.03* | -.32* | .08* | - | | |
| 19. NoClub | .10±.43 | .17* | -.07* | -.05* | -.12* | -.13* | -.10* | -.07* | -.02* | .06* | -.13* | .00 | .00 | .00 | .00 | .03* | .07* | -.13* | -.04* | - | |
| 20. Trust | 3.5±.75 | .04* | .10* | -.11* | -.18* | -.14* | -.10* | -.09* | -.03* | .08* | -.15* | -.10* | .19* | .19* | .20* | .12* | .21* | -.16* | -.12* | .00 | - |

*Note*: M = mean. SD = standard deviation. MTS (moderate equivalent minutes traditional sport per week), BMI (body mass index; kg/m$^2$), HeStat (health status), IM (intrinsic regulation), IDM (identified regulation), IJM (introjected regulation), EM (external regulation), AM (amotivation), Capab (capability), Happy (happiness), SelEff (self-efficacy), LifeSat (life satisfaction), (Lonely) loneliness, IMD (2019 Index of Multiple Deprivation), Opp (opportunity to participate), CNI (community needs index), NoClub (number of clubs). Sig ($p = < .001^*$)

[14, 25, 26, 32–34] is limited in the extent to that it explains the role of correlates across regions. Adjusted data indicated older adults participate in a moderate volume of minutes of traditional sport per-week (43.6±47.43), or 29% of the recommended weekly physical activity guidelines [8]. There is good evidence that moderate levels of sports participation can improve health outcomes [4, 23–27], and meaningfully reduce mortality risk [51, 52]. Our analysis indicates sports participation in England, is associated with age and multiple deprivation on the individual-level, and perceptions of available opportunity on the regional-level. However, unlike previous research [25, 26, 29–31], we did not find other intra-personal correlates, such as gender, body-mass index, and disability status to underpin behaviour meaningfully.

The inverse relationship observed between age and MEMS of sports participation across England is consistent with a body of research [14, 21, 25, 32–34], that has stressed the importance of Active Partnerships, deliverers and providers, the third sector, and sports delivery stakeholders (e.g., national governing bodies) adapting facilities, programmes, and modalities of sport (e.g., modifying coaching and delivery style) for the ageing population [21]. Where such changes to the delivery or modality of sports provision have been undertaken, they are often designed to enhance participants' perceptions of competence in the activity (capability) or promote reasons to engage in the activity (motivation). Contrary to previous qualitative research [31, 40], our results showed neither capability nor motivation to be important correlates of sports participation in older adults. This perhaps indicates that adaptable changes in sports modality may need to go beyond capability or motivation, and stem into the changing

**Table 3. Multilevel model predicting sports participation.**

| Variable | Model | ICC | -2ll ($^{df}$p-value) | $u_{0j}$ | $u_{2j}$ | $e_{0ij}$ | $\beta$(SE) | p-value |
|---|---|---|---|---|---|---|---|---|
| | Null Model | .82 | 907596.70 ($^{df=3}$ p = .001) | 81734.18 | - | 17669.63 | 155.10 (2.30) | - |
| | Model 1 | - | 14559.82 ($^{df=19}$ p = .001) | 35362.87 | - | 654.85 | - | - |
| | Model 2 | - | 14261.14 ($^{df=20}$ p = .001) | 28748.93 | - 11610.88 | 679.60 | - | - |
| Intercept (constant) | | | | | | | 43.64 (7.43) | .001 |
| Age | | | | | | | -.24 (.14) | .040 |
| Gender (*female*) | | | | | | | -1.05 (1.68) | .275 |
| Gender (*other*) | | | | | | | 10.28 (53.64) | .421 |
| Disability (*No disability*) | | | | | | | -2.49 (2.79) | .180 |
| Body mass index | | | | | | | -.001 (.196) | .950 |
| Index of multiple deprivation | | | | | | | .70 (.39) | .030 |
| Capability | | | | | | | -1.69 (1.64) | .150 |
| Opportunity | | | | | | | -28.70 (4.37) | .001 |
| Intrinsic motivation | | | | | | | 1.35 (1.83) | .241 |
| Identified motivation | | | | | | | 2.63 (2.03) | .092 |
| Amotivation | | | | | | | -.61 (1.17) | .301 |
| Health status | | | | | | | -.37 (1.41) | .393 |
| Happiness | | | | | | | -.81 (.61) | .093 |
| Self-efficacy | | | | | | | -1.29 (1.20) | .140 |
| Loneliness | | | | | | | -.01 (.95) | .491 |
| Community needs score | | | | | | | .001 (.001) | .970 |

Note: ICC = interclass correlation. -2ll = -2*loglikelihood (deviance in IGLS estimation). Active Partnership level ($u_{0j}$ = intercept and $u_{2j}$ = slope). Individual level ($e_{0ij}$ = intercept).

lifestyles (e.g., retirement), roles within society and interests of older adults (i.e., opportunity) [33, 34]. This may be understood by adopting participatory methodologies implemented in places, spaces, and environments. These methods should understand the contextual differences in the needs, experiences, and attitudes of older adults participating and not participating in sport.

Further, consistent with research [53–55], the present analysis found that indices of multiple deprivation are inversely associated with sports participation. Barriers to participation in sport or physical activity stemming from deprivation include the provision of and access to facilities and public transport, reduced disposable income, health inequalities, and crime [53–55], factors likely exacerbated during the COVID-19 pandemic [12]. In addressing these societal challenges to accessing opportunity, Sport England (i.e., Uniting the Movement) and UK Government (i.e., Get Active) strategies place emphasis on the importance of place-based approaches to target those most at need and whole-systems models which encourage collective working (e.g., local delivery pilots). This to some extent suggests that the policies and strategies adopted by Sport England may need to go further to reach individuals facing the greatest levels of deprivation within society. Though due to its cross-sectional design, the present analysis can provide little conclusive evidence of the impact of these current policies, it does further reinforce the importance of providing interventions and provisions of sport that can reach, engage, and sustain participation of older adults in the most deprived areas.

Finally, our analysis indicates that the variation observed in sports participation across Active Partnerships in England is attributable to differences in perceptions of opportunity. However, the extent in which this is due to social (e.g., people to play sport with) and/or physical (e.g., facilities to play in, natural space) opportunity [35] cannot be determined from the

data collected within the Active Lives [34, 39] survey. Our analysis is consistent with a body of qualitative research [30, 31, 40], reviews [21, 23, 25], programme evaluations [27] and a cross-sectional study [29] within individual sports which suggest that social opportunity is enriched by a presence of a regional or local cohesive, socially resourced (e.g., competitive and social groups), and supportive club environment. These environments, spaces, and places can be used to promote basic psychological needs for relatedness, social identity, and longevity in sport [21, 56]. In both the case of social and physical opportunity, these studies have noted the importance of 'awareness' of opportunities, and the extent in which this cognizance mirrors the absence or indeed presence of objective opportunities within a given region or local authority [23, 25, 27, 29–31, 40]. Further, these studies [23, 25, 27, 29–31, 40] have stressed the importance of physical and natural environment. This modifiable factor includes suitable inclusive age-appropriate sessions, free to access natural space, and equipment to support participation, as well as ensuring facilities are available and accessible to older adults [23, 25, 27, 29–31, 40]. Understanding how, when, and where older adults experience and engage with sport across a variety of domains is a useful area for future qualitative research.

The findings of the current study are consistent with a cross-sectional analysis of Active Lives data conducted throughout the COVID-19 pandemic [57], in that opportunity was associated with recreational sports participation. Research has indicated opportunity, whether social or physical to be the single greatest predictor of recreational participation in sport or exercise [31, 35, 57]. More specifically, though each factor within COM-B can independently predict behaviour [35], opportunity perhaps is where the greatest changes in population level behaviour can be leveraged [31, 35, 57]. For example, increasing physical (e.g., through the provision of inclusive facilities) and social (e.g., raising awareness of club, supportive networks) opportunity can modify perceptions of capability, and redirect reflective motivation [57]. In the present study, opportunity was strongly associated with capability on the individual-level. In the case of sports participation, this may mean the provision of inclusive, accessible, and age-appropriate facilities can modify perceptions of capability (e.g., improved perceptions of competence, self-efficacy), or an individual's capability informs their perceptions of which opportunities are available or accessible to them [35, 58].

Changing perceptions of opportunity does not however happen by chance [57]. Opportunity can be influenced by the interventions, programmes, strategies, and policies of regional- and national-stakeholders [57] such as Active Partnerships, local-government, the third sector, and sports delivery stakeholders, an area where Sport England and UK Government have invested meaningfully from a policy and fiscal perspective in the past 10-years [20]. In the past decade, Governmental and Sport England policies and strategies have transitioned from supporting programmes with the intention of improving key health outcomes (e.g., mental wellbeing) for individuals and groups in defined settings via simplistic behaviour models (e.g., Towards an Active Nation; 2016–2021), to an approach that encourages systems-based working practices within regions (i.e., Uniting the Movement; 2021-ongoing). This later approach has the capacity to connect communities (e.g., social opportunity), respond to local needs, and modify environmental factors (e.g., physical opportunity) via encouraging working in partnership with regional policymakers, stakeholders and deliverers who are involved in sport, and related sectors connected to its promotion (e.g., healthcare, local-government). However, the legacy of a decade of policy that focused less on regional-level needs, and more on addressing national level key performance indicators (e.g., promotion of health) via short-term programmes should not be overshadowed, in terms of its ability to address inequalities in opportunity across regions of England.

Further, given our findings, it is logical to suggest these bodies must do more to provide, promote or raise awareness of opportunities within regions. Logical steps may include national

policymakers and regional stakeholders (i) identifying where challenges to opportunity exist, (ii) how opportunity varies across people, and (iii) how to best modify opportunities across regions. These recommendations may be achieved through the systems-based or placed-based approaches adopted by both Sport England, Active Partnerships and across the integrated care systems and partnerships [42, 59–61]. From an evaluation perspective, researchers may seek to understand how policy implementation varies across regions, and the extent to which, this represents the provision of built facilities or facilitation of settings that promote socially-rich opportunities.

Promoting opportunity to participate in socially-enriching sports provision (e.g., clubs, groups) and providing facilities, spaces, and places for sports participation to occur are also logical steps for partnerships to adopt. COM-B and its behaviour change wheel suggest logical intervention functions to leverage change in opportunity [35]. These include environment restructuring, such as changing social prospects, facilities, clubs, spaces, and places, and enablement, through increasing enablers and reducing barriers to participation beyond simple restructuring [35]. These can be supported through changes to policy such as guidelines, regulation, legislation, and fiscal support on a regional- and national-level, and environmental and social planning, and changes to service provision on the local-level [35].

## Considerations, limitations, and future directions

A strength of the present study was the analysis of a population-level dataset. The Active Lives data is the only dataset of physical activity, sport, and exercise participation that is representative of the population within England. This analysis therefore provided needed clarity of what determines sports participation in a seldom understood population and provides insight into the role of these factors across Active Partnerships. However, several methodological considerations relating to the Active Lives survey should be noted alongside our findings. Foremost, the current study presents only a single cross-sectional timepoint of the Active Lives survey. Secondly, the extent to which participation is influenced by each factor within COM-B (e.g., physical and psychological capability) and each psychosocial dimension is limited by the single items adopted within the Active Lives survey.

Importantly, these data provided a snapshot during the COVID-19 pandemic. Whilst the Active Lives survey does collect data over relatively equally-weighted waves, it is plausible correlates of behaviour (particularly opportunity) may have differed to a pre-pandemic state, or have been influenced on a situational (e.g., daily) wave-by-wave basis as a response to the pandemic within regions. For example, opportunity to participate and as such participation may have been influenced by home-quarantine and self-isolation [15], closure or adaption of facilities [16, 17], and regional and national mitigation strategies [18]. Further, one analysis of Active Lives data by Strain and Colleagues [14], conducted within the first phase of the pandemic (March-May 2020), found meaningfully reduced participation across all modes of physical activity. However, initial unadjusted analysis of November 2021 data [8] indicates, whilst not at pre-pandemic levels, participation in all modes of sport and leisure activity is increasing.

To any extent, given the economic, societal, and political changes and challenges ongoing as the UK transitions into an endemic phase (e.g., reduced public spending) and the likelihood COVID-19 will not be last pandemic encountered [13], our findings remain important in that they highlight the most recent snapshot in the data. Our findings stress the importance of providing and promoting opportunity as a response to the regional challenges created by the pandemic. A recent call for action underscores the need for adopting systems-based approaches to pandemic preparedness for maintaining healthy living [12, 16]. Approaches such as this include the creation of policy which creates sustainable, free to access facilities and resources

in times of a pandemic, therefore altering perceptions of opportunity and encouraging participation [16].

Given such regional- and national-policy (e.g., United the Movement, COVID-19) shapes correlates of over time, future research may consider adopting a multi-stage cross-sectional study approach to our analysis [14]. Moreover, the present analysis sought to examine variation in sports participation across Active Partnerships as a function of opportunity. Whilst, in most cases, older adults would be likely to participate within the regional partnership (i.e., typically a county), it cannot be excluded that some participation would occur outside of the regional partnership, for example, as a response to the absence of perceived opportunity or living close to a partnership border. The circumstances described are unavoidable within the analysis. Further, due to scope of the survey aiming to understand all modes of physical activity, each item despite referencing to sports, fitness, and recreational activities (through an asterisk) is worded to refer to 'physical activity' rather than sport directly. Finally, whilst providing a strong insight into where variation exists within sports participation and what determines this variation, these data do not have the capacity to explain how, why, and when variation is occurring. Given the emphasis of variation in sports participation between regions, a participatory approach that focuses on regional-level provision may be most appropriate. Current efforts utilising systems-thinking methodologies with stakeholders [42, 59–61], may be a promising approach to understand how opportunity is created and modified within regions.

## Conclusion

The current study presents the first analysis of Active Lives sport participation data in older adults across England. Consistent with analysis of Active Lives data [14, 32–34] and systematic reviews of sports participation [21, 25] age and deprivation inversely predicted participation. Further, our analysis suggests across regions perceptions of opportunity influence participation in sport. This underscores the importance of nationwide and regional policy that creates socially engaging physical spaces, places, and facilities for older adults to engage within sport. It remains important to monitor our observations longitudinally as changes in policies relating to opportunity are enacted.

## Acknowledgments

The authors of the present study have no competing interests to declare. None of the authors have any financial or relationships to declare which can influence or bias the outcome of the present study. The work presented is an original analysis of an existing dataset.

## Author Contributions

**Conceptualization:** Andrew Brinkley, Gavin Sandercock, Ruth Lowry, Paul Freeman.

**Formal analysis:** Andrew Brinkley, Gavin Sandercock, Paul Freeman.

**Investigation:** Andrew Brinkley.

**Methodology:** Andrew Brinkley, Gavin Sandercock, Paul Freeman.

**Project administration:** Andrew Brinkley.

**Resources:** Gavin Sandercock.

**Validation:** Gavin Sandercock.

**Writing – original draft:** Andrew Brinkley, Gavin Sandercock, Ruth Lowry, Paul Freeman.

**Writing – review & editing:** Andrew Brinkley, Gavin Sandercock, Ruth Lowry, Paul Freeman.

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
