## [Decision Letter · Decision Letter 0]

9 Aug 2023

PONE-D-23-04952What Determines Participation in Sport for Older Adults in England: A Multilevel Analysis of Active Lives DataPLOS ONE

Dear Dr. Adrew Brinkley 

Thank you for submitting your manuscript to PLOS ONE. After careful consideration, we feel that it has merit but does not fully meet PLOS ONE’s publication criteria as it currently stands. Therefore, we invite you to submit a revised version of the manuscript that addresses the points raised during the review process. Please submit your revised manuscript by Sep 23 2023 11:59PM. If you will need more time than this to complete your revisions, please reply to this message or contact the journal office at plosone@plos.org. Please include the following items when submitting your revised manuscript:A rebuttal letter that responds to each point raised by the academic editor and reviewer(s). You should upload this letter as a separate file labeled 'Response to Reviewers'.A marked-up copy of your manuscript that highlights changes made to the original version. You should upload this as a separate file labeled 'Revised Manuscript with Track Changes'.An unmarked version of your revised paper without tracked changes. You should upload this as a separate file labeled 'Manuscript'.

We look forward to receiving your revised manuscript.

Kind regards,

Timoteo Salvador Lucas Daca, Ph.D

Academic Editor

PLOS ONE

Journal Requirements:

The author(s) received no specific funding for this work. The Active Lives survey is a funded project by Sport England. However, the data analysed is publicly available on the UK Data Archive. Therefore, Sport England played no role in the analysis of the data or interpretation of the findings. Sport England was not offered the right to review our methods or analysis.

The author(s) received no specific funding for this work. The Active Lives survey is a funded project by Sport England. However, the data analysed is publicly available on the UK Data Archive. Therefore, Sport England played no role in the analysis of the data or interpretation of the findings. Sport England was not offered the right to review our methods or analysis.

Reviewers' comments:

Reviewer's Responses to Questions

**Comments to the Author**

1. Is the manuscript technically sound, and do the data support the conclusions?

Reviewer #1: Partly

Reviewer #2: Yes

2. Has the statistical analysis been performed appropriately and rigorously? 

Reviewer #1: Yes

Reviewer #2: Yes

3. Have the authors made all data underlying the findings in their manuscript fully available?

Reviewer #1: No

Reviewer #2: Yes

4. Is the manuscript presented in an intelligible fashion and written in standard English?

Reviewer #1: Yes

Reviewer #2: Yes

5. Review Comments to the Author

Reviewer #1: Comments on the paper: What Determines Participation in Sport for Older Adults in England: A Multilevel Analysis of Active Lives Data

This is an interesting idea, to look at the correlates of sport participation large population data set in England. There are some methodological as well as conceptual limitations in the paper that need to be addressed, in my view, to improve the quality and generalisability of the findings herein.

The paper looks at variation in sport participation through the regional active partnerships that occurred in England to promote sport. The study looks at re-analysing the sport England Active Lives survey in 2020-2021. This is any partially theoretically driven paper, examining motivation and opportunity to explain sport participation behaviour, in the context of regional and supra-individual factors, including environments and opportunity. However the paper is also opportunistic in that it creates a partial theoretical framework out of the questions that were asked in the active lives survey, rather than designing a theoretical study a priori. The strengths are the use of a very large population data set, such that micro regional analyses are possible.

A key question is whether people in England are becoming more inactive, which is alluded to in the Background para 1 line 3. One would need trend data to justify that, which should be possible to obtain from active lives analyses at least since around 2016. The regional Active Partnerships occurred as a result of the interest by sport England in investing in physical activity and sport following the London 2012 Olympics. However there is no evidence in this paper whether these partnerships implemented sport promoting activities and environments well, and whether the policies were differentially implemented across different regions; the reason why this is important is because it implies that policy implementation is more than just opportunities in the built environment as they exist, but could be a range of diverse and different strategies across regions. Has that engagement occurred, and has sport participation changed in relation to those regional partnerships? This is a fundamental underpinning concept for this paper which would need both policy analysis and further investigation of additional active lives datasets from other years. Given a decade of this policy implementation, a critical question is whether sport participation has increased, and whether the active partnerships have contributed to that. Otherwise it undersells the relevance of active partnerships in this specific analysis.

The study refers to determinants, but the term correlates should be used throughout. The authors report on previous research of correlates of sport participation and identify issues such as socio-economic status, geography gender opportunities capability and motivation in previous research. The theoretical framework here, COM-B is increasingly used in understanding physical activity participation and is likely to be relevant to sport participation, although this framework may have somewhat less relevance for older adults. Because of the large sample size, and n=66 808 , the study has sufficient power to examine regional variation as well as different age groups. Understanding older adults older than 84 years is important, as their meanings of physical activity and access to and relevance of what they do is quite different to those in the younger age groups; this is partially true for the next youngest age group, those aged 75 to 84, and these limitations, both physical and psychological amongst older elderly are not really dealt with well in this paper.

There is no rationale for the cutpoint of 3360 minutes per day, which is eight hours per day, which is quite unrealistic as an upper limit for truncation or winsorisation, A even though it is referenced in reference number 33. Further thought on the nature of realistic cutpoints for older adults should be considered. There is good classification of sports into those that are moderate and vigourous, but they should be listed at least in supplementary online material. One question regarding the 28 day prevalence in the active lives survey is whether seasonal variation might not influence this, or is the active lives survey a rolling continuous data collection?

The measures from the active living survey will describe, but are often single item is used here to reflect call underlying psychological and psychosocial dimensions. It would be useful to have some psychometric work undertaken on these data, rather than just referencing the published scales or other work on the dimensions from which these items were developed.

The multilevel analysis seems appropriate, but is often quite complicated, such as in table 2 with the correlation matrix of 20 variables. Given the very large sample size although most of the correlations are <0.1, they are all highly significant, an artefact of sample size .

The analysis describes the average moderate equivalent minutes as being 68 minutes per week for sport participation. First, the variance around this estimate is extremely high, being at least 3 1/2 times the median value. Second, is this a realistic meaning for older adults? It depends on the kinds of activities that are included in sport. For example if this is organised sport, then this is a non-credible mean value for the population older adults, but if it includes walking it could be plausible as an estimate. Given the high variance, most regions have very large lower and upper bound limits, and this does somewhat attenuate regional differences.

The analysis regional differences.identifies factors associated with sport participation, after adjusting for variation across the regional active partnerships.

a small comment , and I have not assessed the paper for typographical errors generally, but the Isle of Wright should be "Wight".

figure 1, the heat map shows that data in north-western England in Cumbria report more sport minutes, as do other regions in Northumberland and Yorkshire. The inverse is true of the south-east, where five fewer minutes are reported in Kent, Sussex, Cambridge or Hampshire. This is the inverse of physical activity that usually shows a socio-economic gradient where the south and south-east of England are more active than the more socially disadvantaged North, and reasons why these data shows such unexpected inverse gradients by deprivation should be explored.

Table 3 accounts for regional variation and assesses the correlates of sport, finding that factors such as age are strongly correlated. Surprisingly, and contrary to the majority of published research, gender differences in sport participation, differences by disability status and differences by body mass index appear non-significant. These unexpected correlates, or lack thereof require careful explanation in either the model all measures used. This could also explain why capability or motivation are not significant here, as in this could be a measurement issue.

On page 16 the authors mention the Covid19 influence on the results. This is important, as Covid undoubtedly influenced both individual and environment level correlates, capacities for sport participation and individual motivation for being active. It would be important to re-analyse and present data for another year, ideally pre Covid, to assess the same correlates persist . This is really important in terms of the generalisability of these correlates, as physical activities that people could do and sports that people could participate in were very different during the pandemic. As shown by Strain et al 2022 in the reference list, analyses of the active lives data over five years shows that physical activity and sport declined especially driven by 2020 restrictions. Since these restrictions overlap with the period study in this analysis, it is really important to see whether these correlates remained consistent at other time points .

Finally, the policy relevance and the innovation of these findings is not clearly discussed. The lack of gender or disability as correlates, and the presence of regional variation need further exploration and further reanalysis as suggested above. If carried out, a more comprehensive understanding of sport participation would be possibly gleaned from these active lives data.

Reviewer #2: An interesting study, with potential. There are areas for improvement, however.

1. It needs to be made clear in several places that this study is specific to England.

2. A Bonferroni adjusted p value is advised given the sample size and number of tests being run.

3. The COVID 19 pandemic is reduced to a limitation at the end. This is important contextual information and needs to be given much more attention within the manuscript.

4. There are a lot of grammatical errors throughout and re-wording needed in many places. Please undertake a thorough proof read.

Please provide line numbers in future. Otherwise, it makes a reviewer’s job much harder.

ABSTRACT

Could a definition/description of “Active Partnerships” be offered in the abstract, especially to aid an international reader?

The abstract needs to make the context of the dataset clear. At present, it’s not clear which jurisdiction the dataset is drawn from.

There are multiple typing and punctuation errors in the abstract – please proof read thoroughly (determinants, addresses, missing comma after this in L5). There are also grammatical errors (e.g., Our findings indicate THAT when….).

More information is needed on the dataset. For example, how large was the sample? Likewise, these data were collected during the COVID-19 pandemic. Some recognition of this is needed especially as this is likely to have impacted the findings. For example, older adults, the group the authors pick out later, were more impacted by lockdown restrictions than other groups.

It’s not clear until towards the end that older adults are the focus. This needs to be clear from the outset.

Some switching throughout the manuscript between aging and ageing.

MAIN TEXT

P3, para 2: is the “remain” needed? This suggests that if all these issues are addressed, SE will have no responsibly for this.

P3: “which envisaged to “inspire a generation” could benefit from rewording and consideration of grammar rules for punctuation of which. See which again on P4.

P3, comma needed after movement.

P4, Para 2, again a comma missing – after “long-term”. Please undertake a thorough read of the document to catch these.

P4, Para 2, at the end of this paragraph, I am wondering: what is the key message for this paragraph? A summary sentence would be useful.

P4, Para 3: determinants being re-stated is repetitive. I also could not help but feel that “and is” is needed to break this sentence up.

I also suggest there is some rewording needed in how the COM-B elements are described. The first two come across as categorical variables based on how they are worded, whereas the third (which I prefer) suggests it’s continuous (i.e., “extent”).

“Though influenced by social and environmental factors such as an extrinsic stimulus capability and motivation to be individual-level predictors of behaviour(27). Whilst, opportunity varies according to social support, subjective norms, and environmental factors such as the availability of facilities and/or social actors to support participation(27)”. These are not full sentences at present and could benefit from grammar checks.

P5: “As such, opportunity can, in theory, be modulated to the greatest extent by Active Partnerships”. This seems like quite a strong claim here. Couldn’t there be other policymakers with roles that are important too? A tempering of the language here would be beneficial.

P5: I find it hard to agree that exergaming is a form of sport (see definitions of sport elsewhere; dictionary of sport etc.). Can the authors justify this?

P5; in the middle of the page, which should be that.

P5: should not-population be non-population?

P5: should “small” be included before qualitative or something like this? Otherwise, a hierarchy of evidence is being set.

Aim paragraph: similar to the abstract, please make the geographic location of this work clear.

“These data report during the COVID-19 pandemic” needs rewording.

P6, should “miss coded” be “miscoded”

P6, why was 60 selected as the marker of older adult? In the introduction, 55 was the lower end of the range cited.

P7, explain who IPSOS MORI are (within parenthesis even) for an international reader.

P7, the definition of sport provided here comes too late. It needs to be in the introduction.

P7, hyphens are necessary for 28-days and 4-weeks (four weeks).

P7: as currently written “The Active Lives survey requested participants to provide data on the number of days they participated in 200 modes” gives the impression they had to have participated in all 200 forms are currently worded. Re-wording needed.

P7, commas needed before and after “including” and Archive) in the middle of this page when referring to the survey.

P7, “Moderate Equivalent Minutes (MEMS) spent participating in traditional modes of sport was extracted from the data.” Review whether plural or singular are used correctly here.

P7, “MEMS traditional team sport is constructed of activities representing moderate (MITS) and vigorous intensity traditional sports (VITS) into a composite variable.” Grammar seems off here. Please reword for reader.

The word “construct” confused me and I still don’t fully understand the meaning here. Overall, the explanation from “For the purpose of the present study” to the end of this paragraph could be much better. A link could also be provided direct to the UK Data Archive page.

P8, there is a parenthesis within a parenthesis unnecessarily in the middle of the page.

P8, it seems somewhat of a stretch to suggest the question for “individual development” is either reflective of psychological wellbeing or individual development. It seems to be a measure, to some degree, of generalised self-efficacy.

P8, should active partnerships have capital letters near the bottom?

P9; no need for Foremost. Do you mean first?

One thing I would ask about is were the information regarding when the data were collected? It seems important that some data were collected during lockdown periods. Was this controlled for?

Given the sample size and the number of variables, a Bonferroni adjusted p value should be used. The correlation table is showing that even r = .02 is significant, yet that is really a negligible relationship.

Table 1: the n for each Active Partnership would be useful.

Table 3: some p values are 2 decimals, and others 3.

P15: “national population representative picture” either needs punctuation or needs to be more concise. This sentence also needs to state the location. The English focus of the study also needs to come through in subsequent sentences.

P15: “29% of the recommended weekly guidelines” – be clear that these are recommended PA guidelines, not sport.

P15: the data are cross-sectional, so the idea of prediction cannot be substantiated by the dataset.

P15: the “, which” in the second line of para 2 should be that.

P15: this finding that COM-B elements are not working is consistent with lots of research in exercise psychology pointing to the limits of such social-cognitive models (see Ekkekakis).

P16, comma before which needed.

While it is being suggested that better sport options are needed, do older adults actually want sport in these areas? There is great room here to point towards the need to work with older adults in these communities to establish what works.

P16: “and places have been associated with the basic need relatedness, belonging, an identity, and longevity in sport” – this needs to be reworded.

P16: unnecessary parenthesis after accessible.

P16, “whether social or physical”- a comma is needed before and after this segment.

P17, comma needed before “An area”.

P17: should perceptions of opportunity be modified solely? What about changing actual opportunity? This comes later, but surely should be coming first.

P17: “participatory placed based approaches” needs several pieces of punctuation.

P17: socially enriching should be “socially-enriching,”…

6. PLOS authors have the option to publish the peer review history of their article (what does this mean?). If published, this will include your full peer review and any attached files.

Reviewer #1: No

Reviewer #2: No

---

## [Author Response · Author response to Decision Letter 0]

9 Oct 2023

Please see the attached document.

---

## [Decision Letter · Decision Letter 1]

22 Feb 2024

PONE-D-23-04952R1What determines participation in sport for older adults in England: A multilevel analysis of active lives dataPLOS ONE

Dear Dr. Brinkley, Thank you for submitting your manuscript to PLOS ONE. After careful consideration, we feel that it has merit but does not fully meet PLOS ONE’s publication criteria as it currently stands. Therefore, we invite you to submit a revised version of the manuscript that addresses the points raised during the review process. **In points two and three, the reviewers do not fully agree with yours arguments. I suggest focusing on these points to find a balance according to each reviewer's comments. At points where there are conflicts between reviewers, I ask to follow the instructions of the reviewer #3. There are many improvements on the manuscripts that need minor revision. ** Please submit your revised manuscript by Apr 07 2024 11:59PM. If you will need more time than this to complete your revisions, please reply to this message or contact the journal office at plosone@plos.org. Please include the following items when submitting your revised manuscript:A rebuttal letter that responds to each point raised by the academic editor and reviewer(s). You should upload this letter as a separate file labeled 'Response to Reviewers'.A marked-up copy of your manuscript that highlights changes made to the original version. You should upload this as a separate file labeled 'Revised Manuscript with Track Changes'.An unmarked version of your revised paper without tracked changes. You should upload this as a separate file labeled 'Manuscript'.If applicable, we recommend that you deposit your laboratory protocols in protocols.io to enhance the reproducibility of your results. Protocols.io assigns your protocol its own identifier (DOI) so that it can be cited independently in the future. For instructions see: https://journals.plos.org/plosone/s/submission-guidelines#loc-laboratory-protocols. Additionally, PLOS ONE offers an option for publishing peer-reviewed Lab Protocol articles, which describe protocols hosted on protocols.io. Read more information on sharing protocols at https://plos.org/protocols?utm_medium=editorial-email&utm_source=authorletters&utm_campaign=protocols.

We look forward to receiving your revised manuscript.

Kind regards,

Timoteo Salvador Lucas Daca, Ph.D

Academic Editor

PLOS ONE

Journal Requirements:

Reviewers' comments:

Reviewer's Responses to Questions

**Comments to the Author**

1. If the authors have adequately addressed your comments raised in a previous round of review and you feel that this manuscript is now acceptable for publication, you may indicate that here to bypass the “Comments to the Author” section, enter your conflict of interest statement in the “Confidential to Editor” section, and submit your "Accept" recommendation.

Reviewer #2: (No Response)

Reviewer #3: All comments have been addressed

2. Is the manuscript technically sound, and do the data support the conclusions?

Reviewer #2: Yes

Reviewer #3: Partly

3. Has the statistical analysis been performed appropriately and rigorously? 

Reviewer #2: Yes

Reviewer #3: No

4. Have the authors made all data underlying the findings in their manuscript fully available?

Reviewer #2: Yes

Reviewer #3: Yes

5. Is the manuscript presented in an intelligible fashion and written in standard English?

Reviewer #2: Yes

Reviewer #3: Yes

6. Review Comments to the Author

Reviewer #2: A much improved manuscript. I only have a few minor comments to polish the manuscript now.

Title: Should there be capitals for Active Lives? I also suggest saying England.

P4, L84 and L96: that instead of which.

Discussion

I suggest a thorough check through here to ensure you are writing in the past tense. For example, P19, L4 and 8 should be indicated. Please check the reminder. Further examples on P20 (are should be were).

P19, L26: remove the comma before that.

P20, L55: comma before which.

P22, L91: do you mean simple rather than simplest?

P22, L94: remove hyphen between local and needs.

P22, L97: that instead of which. Same on p22, L110.

P23, L133: remove hyphen between single and items.

P23, L137: remove comma and add hyphen between equally and weighted (I think).

P23, L143-144: is a “that” missing here?

L147: these data.

Reviewer #3: Manuscript Number: PONE-D-23-04952R1

Manuscript Title: What determines participation in sport for older adults in England: A multilevel analysis of active lives data

The authors conducted a cross-sectional study to explore the levels of sport participation in older adults, the correlated of sport participation in older adults, and the differences in sport participation across Active Partnerships in older adults. The study found that age, socioeconomic status, and perceptions of opportunity were associated with sports participation in older adults. Strengths of the study included the large representative sample. However, there were limitations including the cross-sectional design, self-report measures, and the fact that the data was collected during the pandemic may limit the generalisability. The authors have done a good job at addressing the reviewer comments. However, I have some additional comments that should be addressed before publication.

Major comment: Reviewer 2 said that COVID 19 pandemic is reduced to a limitation at the end. This is important contextual information and needs to be given much more attention within the manuscript. I can see that the authors have added more details around this, but I think it needs to be emphasised even more, especially in the introduction. Given the data was possibly collected during the lockdowns, these need to be discussed. What impacts did covid and the lockdowns have on sports organisations and individuals levels of sport participation? Sport participation is likely much lower for all people during this time, and the associations between independent variables and sport participation may be different during this time. Can these findings be generalised to post pandemic times?

Abstract

Line 28: What do the authors mean by “Sports participation forms a bedrock of policy”. This is unclear and is somewhat disjointed from the first sentence.

The authors should clearly state the primary and secondary aims of the study.

Lines 38-39 “Further, sports participation differed significantly across regions due to changes in the perceptions of opportunity to participate (β=-28.70, p=.001).” Please reword, the multilevel model conducted does not allow these causal statements.

Background

Lines 44-45: The authors could provide evidence for the specific health benefits of physical activity for older adults. For example, the benefits of physical activity for older adults includes fall prevention, improved prevention and management of non-communicable diseases, improvements in mental health, cognitive health, and strengthened social connections.

Lines 65-66: The final sentence of the first paragraph reads a little disjointed.

A stronger justification for using the COM-B model is needed.

Lines 153-154 The authors state that “Importantly, research has also not considered how correlates of behaviour interact across regions of a nationwide population”. The term interact has a specific statistical meaning, and this sentence could be misleading as the authors of this paper did not look at the interactions between correlates.

Please provide some contextual information about sport participation during the covid pandemic.

Methods

Line 263: should say sequentially.

Line 267: What do the authors mean by “a significant prediction within general linear regression models.” Please clarify. It sounds like the authors have conducted univariate general linear regression models to determine which variables were included in the final model. If this is the case, please provide the results of these univariate models, possibly as supplementary material.

Please justify why both a random intercept and random slopes model were used.

Line 276 – “(model 2) was constructed where sports participation could vary as a function of change in opportunity across regions.” This isn’t quite accurate. The random slope allows the relationship between explanatory variables and sport participation to vary across regions. Please also check the wording around the random intercepts model.

Did the authors consider building the model sequentially, so adding the individual level correlates first, and then adding in the Active Partnership variables? It is likely that there would be evidence for some of the individual level correlates, such as gender, before adjusting for the Active Partnership variables. This would allow the authors to assess the extent of confounding and provide a more comprehensive understanding of the associations between the explanatory variables and sport participation.

Results

The authors report variance explained at different levels throughout the results, for example “this model explained 53% of the variance on the Active Partnership level”. Firstly, is this an R2 statistic, please clarify. Typically, R2 statistics indicate the proportion of variation in the outcome explained by all of the variables in the model, not the level. Please check and provide further details.

Line 299: “.. significantly fitted the data” Please reword to good fit, rather than significant fit.

Lines 301-302 “More specifically, as opportunity was perceived as more favourable, sports participation increased meaningfully (β=-28.70, p=.001).” Sports participation increased and the negative coefficient may be confusing. Please consider rewording this.

Are the coefficients standardised or unstandardised? Please clarify.

Table 3. This is difficult to read. The authors could consider providing the model fit statistics in a separate table, a table footnote, or within the text.

Discussion

Lines 40-47: The paragraph on the association between deprivation and sport participation needs more discussion. What has been done to reduce the gap to date? What are the barriers that these people face? How can sports engage older adults in deprived areas?

The authors do a good job at describing the limitation to their study. However, I still believe that the context of COVID is a major limitation, please expand on why this is a limitation and the impact it has on the generalisability of the results.

Lines 171-172: The authors conclude that “Further, perceptions of opportunity differ between regions, and this appears to modify participation meaningfully.” The authors should be careful with their language, the findings from this study do not support this bold claim. The authors did not conduct moderation analyses. Further, this is a cross-sectional study; the authors should not claim that the results from this study suggest that perceptions can actually change behaviour.

7. PLOS authors have the option to publish the peer review history of their article (what does this mean?). If published, this will include your full peer review and any attached files.

Reviewer #2: No

Reviewer #3: No

---

## [Editor Report · Decision Letter 2]

24 Mar 2024

What determines participation in sport for older adults in England: A multilevel analysis of Active Lives data

PONE-D-23-04952R2

Dear Dr. Andrew Brinkley,

We’re pleased to inform you that your manuscript has been judged scientifically suitable for publication and will be formally accepted for publication once it meets all outstanding technical requirements.

Kind regards,

Timoteo Salvador Lucas Daca, Ph.D

Academic Editor

PLOS ONE
---

## [Editor Report · Acceptance letter]

26 Mar 2024

PONE-D-23-04952R2 

PLOS ONE

Dear Dr. Brinkley, 

I'm pleased to inform you that your manuscript has been deemed suitable for publication in PLOS ONE. Congratulations! Your manuscript is now being handed over to our production team.

Kind regards, 

on behalf of

Dr. Timoteo Salvador Lucas Daca 

Academic Editor

PLOS ONE